# Dual Disorders in the Consultation Liaison Addiction Service: Gender Perspective and Quality of Life

**DOI:** 10.3390/jcm10235572

**Published:** 2021-11-26

**Authors:** Teresa Ferrer-Farré, Fernando Dinamarca, Joan Ignasi Mestre-Pintó, Francina Fonseca, Marta Torrens

**Affiliations:** 1Department of Experimental and Health Sciences (CEXS), Universitat Pompeu Fabra, 08002 Barcelona, Spain; teresaff95@gmail.com (T.F.-F.); jmestre@imim.es (J.I.M.-P.); 2Department of Psychiatry and Legal Medicine, Universitat Autònoma de Barcelona, 08290 Cerdanyola del Vallès, Spain; mtorrens@parcdesalutmar.cat; 3Institute of Neuropsychiatry and Addictions, Parc de Salut Mar, 08003 Barcelona, Spain; fdinamarca@psmar.cat; 4Addiction Research Group, IMIM-Institut Hospital del Mar d’Investigacions Mèdiques, 08003 Barcelona, Spain

**Keywords:** dual diagnosis, substance use disorders, consultation liaison service, quality of life, gender

## Abstract

Dual disorders (DD) and gender differences comprise an area of considerable concern in patients with substance use disorder (SUD). This study aims to describe the presence of DD among patients with SUD admitted to a general hospital and attended by a consultation liaison addiction service (CLAS), in addition to assessing its association with addiction severity and quality of life from a gender perspective, between 1 January and 30 September 2020. The dual diagnosis screening interview (DDSI), the severity of dependence scale (SDS), and the WHO well-being index were used to evaluate the patients. In the overall sample, DD prevalence was 36.8%, (women: 53.8% vs. men: 32.7%, NS). In both genders the most prevalent DD was depression (33.8%, women: 46.2% vs. men: 30.9%, *p* = 0.296). Women presented more panic disorders (46.2% vs. 12.7%, *p* = 0.019) and generalized anxiety (38.5% vs. 10.9%, *p* = 0.049) than men. When DD was present, women had worse quality of life than men (21.7 vs. 50 points, *p* = 0.02). During lockdown period 77 patients were attended to and 13 had COVID-19 infection, with no differences in relation to sociodemographic and consumption history variables. The study confirms a high prevalence of DD among patients with SUD admitted to a general hospital for any pathology, and its being associated with worse quality of life, particularly in women.

## 1. Introduction

Dual disorder (DD) is the coexistence in the same patient of a substance use disorder (SUD) and another psychiatric condition [1]. Whilst not a new phenomenon, it is gaining importance due to its marked prevalence and complexity with respect to the clinical approach of such patients. 

Several studies show that, compared with patients with only SUD, DD patients require a greater number of emergency room admissions and hospitalizations in psychiatry services. They also present higher suicide rates and more risky behaviour associated with mortality and infectious diseases, such as HIV and hepatitis viruses [2,3]. Moreover, in addition to more frequent episodes of violent behaviour, such patients have greater social problems (higher unemployment rates) [4]. Consequently, DD patients present a greater risk of addiction chronicity and severity, their treatment is more difficult and expensive, and they have a worse prognosis than those with only one psychiatric disorder (SUD or other) [5,6]. 

It has also been observed that DD patients have a worse perceived quality of life (QoL) than those with only SUD, a little-studied parameter that is gaining relevance as an indicator of the results of the treatments offered [7,8,9,10].

To date, studies carried out to determine DD prevalence in mental health units and addiction services have reported a high incidence in both cases [11,12,13]. To the best of our knowledge, however, no studies have been performed analysing this prevalence in patients admitted to a general hospital, beyond the emergency room, for any health reason besides SUD. Furthermore, in recent years, interest in gender perspective in the study of addictions has increased [14]. Gender plays a crucial role in determining vulnerability, clinical presentation, and treatment outcomes in patients with SUD. Women are more vulnerable than men in the addiction process, since they progress more quickly from the first substance contact to their addiction (telescoping effect), requiring less dose and time of use to reach a greater degree of addiction severity [14,15,16]. Women with SUD present more medical and psychiatric comorbidities than their male counterparts [17]. In women with SUD (compared to men with SUD) a higher prevalence of infections (HIV, HCV, etc.) has been observed, and in terms of DD, the most common psychiatric disorders are depression, anxiety, and post-traumatic stress disorders (PTSD). Finally, a higher incidence of gender-based violence and history of sexual abuse has been detected among women with SUD, leading them to being more susceptible to psychiatric illness and a resulting worse perceived QoL than SUD men [18,19,20].

The objective of the present study was to analyse DD prevalence among patients with SUD admitted to a general hospital for any health problem, whether related to their addiction or not. They were attended by a consultation liaison addiction service (CLAS), which assessed addiction severity and perceived QoL in addition to a gender perspective. The study was interrupted by the COVID-19 lockdown; consequently, as a secondary objective, we compared the characteristics of patients attending CLAS during those months who were unable to receive face-to-face interview assessment. 

## 2. Materials and Methods

### 2.1. Participants

The study sample was made up of patients with an SUD diagnosis according to DSM 5 criteria [21] admitted to a general hospital (Hospital del Mar) for any health problem, whether directly related to their addiction or not, and attended by the CLAS. Patient recruitment of patients was carried out between 1 January and 30 September 2020. 

### 2.2. Inclusion and Exclusion Criteria

Inclusion criteria were: (1) patients with any SUD diagnosis admitted and assessed by the CLAS at Hospital del Mar; (2) being over 18 years of age; and (3) speaking/understanding Spanish. Exclusion criteria were: (1) documented mental retardation/moderate-severe neurocognitive impairment, with prior neuropsychological evaluation or assessed in the psychopathological examination and the Spanish version of the Montreal Cognitive Assessment 26 (MOCA 26) test [22] (a score 10–17 indicating moderate neurocognitive deterioration, and less than 10 severe neurocognitive deterioration); (2) acute confusional disorder (according to psychopathological examination); (3) not speaking/understanding Spanish; and (4) the patient’s clinical condition hindering evaluation. If any of the exclusion criteria were transitory (such as delirium or intoxication), the interview was carried out once the condition had improved.

### 2.3. Assessment Instruments

For DD screening, a dual diagnostic screening interview (DDSI) was used [23]. DDSI is an instrument that assesses the following mental conditions: panic disorder, generalized anxiety disorder, specific phobia, social phobia, agoraphobia, depression, dysthymia, mania, psychosis, attention deficit hyperactivity disorder (ADHD), and PTSD. The diagnoses obtained with the DDSI are lifetime psychiatric diagnoses. 

QoL was evaluated with the WHO well-being index [24]. This self-administered tool consists of 5 questions that refer to the physical–emotional state of the patient in the previous 2 weeks. The total score obtained ranges from 0 to 100 points, and the higher the score the greater the well-being.

Addiction severity was measured with the Spanish version of the Severity of Dependence Scale (SDS) [25,26], The self-administered SDS consists of 5 questions referring to substance use in the previous year. The total score ranges from 0 to 15 points, a higher score indicates a greater degree of dependence on the substance in question.

### 2.4. Procedure

The CLAS at Hospital del Mar receives daily requests for interventions with patients who present an SUD concomitantly to their cause of admission. As part of standard procedure, clinical and sociodemographic data are collected in a database and include details of substance use and comorbidities (including serological status of HIV and hepatitis viruses). Patients who met the inclusion criteria were informed and, if they agreed to participate, an independent researcher conducted the study. DDSI was performed first followed by the WHO Well-being Index and the SDS. The total time of the evaluation was a maximum of 25–30 min.

Due to the COVID-19 lockdown (from 10 March 2020 to 22 June 2020) the independent researcher had no face-to-face access to the patients. Basic sociodemographic and clinical data gathered from the CLAS in this period were analysed.

### 2.5. Analysis of Data

A descriptive analysis of variables was carried out and possible differences by gender from the interviews were analysed. To do so, the mean, median, standard deviation, range, and frequency were calculated according to the nature of each variable. Kolmogorov–Smirnov normality tests were performed and, consequently, non-parametric tests were used. For the comparison of means between groups the Mann–Whitney and Wilcoxon U tests and Chi-square for categorical variables were employed. All calculations were carried out using the IBM (IBM Corp. Released 2013. IBM SPSS Statistics for Windows, Version 22.0. Armonk, NY, USA: IBM Corp.).

### 2.6. Ethics

Patients who met the inclusion criteria were informed of the characteristics of the study and the confidentiality of data processing. Prior to the interview they were asked to sign an informed consent. There was no impact on the patients’ usual treatment. The study was approved by the Parc de Salut Mar Clinical Research Ethics Committee (SEIC-PSMAR); CEIm project number: 2019/8970/I.

## 3. Results

During the entire period of study, a total of 233 patients were admitted by the CLAS. In the lockdown period (from 10 March 2020 to 22 June 2020) 77 patients were only attended by the liaison team and were included for a separate analysis. Of the 156, a total of 104 were candidates, and 68 patients completed the assessment interviews and were included for the principal objective (see Figure 1). There were no differences in gender proportion or other sociodemographic variables of the included patients versus the non-included ones.

### 3.1. Sociodemographic Characteristics

The characteristics of the 68 patients that completed the interview are shown in Table 1. Women (*n* = 13) represented 19% of the sample. Mean age was 50.99 years (SD *=* 11.67) and there were no differences between genders. Neither were there gender differences found with respect to civil status, employment situation, living conditions, and criminal records (Table 1).

### 3.2. Clinical Characteristics

The clinical characteristics of the 68 patients are shown in Table 2. Regarding the main drug of use, alcohol was present in 47.1% of the patients, followed by opiates (38.2%) and cocaine (10.3%), with no differences by gender. 

The mean commencement age of the main substance was 18.71 years, which was lower in men than women (17.83 vs. 23 years, *p* = 0.018). The mean of total abstinence time was 24.15 months, without differences by gender. More than half the sample (58.9%) had been previously involved in addiction treatment without differences between men and women. There were no differences between genders regarding HIV and hepatitis B and C virus infections (HBV and HCV).

### 3.3. Dual Disorder Assessment

Of the 68 patients interviewed, 25 (36.8%) had positive screening for DD; depression was the most prevalent (33.8%) followed by psychosis and panic (both 19.1%) (Table 3).

Prevalence of DD was 32.7% in men and 53.8% in women. In the former, the most common psychiatric disorder was depression (30.9%), followed by psychosis (16.4%), mania and PTSD (both 14.5%). While in the latter, the most frequent psychiatric disorders were depression and panic (both 46.2%), followed by generalized anxiety (38.5%). 

Panic disorder and generalized anxiety were greater for women than men (*p* = 0.019 *p* = 0.049 for generalized anxiety, respectively) (Table 3).

When analysing DD with other clinical variables, a greater proportion of patients with HIV antibodies was observed (36% vs. 9.3%; *p* = 0.02). No differences were found for the other clinical and sociodemographic variables.

### 3.4. Dual Disorder and Quality of Life

Considering only patients with DD (18 men and 7 women), the QoL index was higher for men ($\bar{x}$:50 points vs. $\bar{x}$:21.7 points, respectively, *p* = 0.02). This difference did not change when the comparison was made excluding patients with HIV antibodies that could bias results. No gender differences were detected in non-DD patients (Table 4).

### 3.5. Dual Disorder and Severity of Addiction

According to SDS, the mean severity of dependence was 6.58 points (SD *=* 4.08) in non-DD patients and 8.16 points (SD *=* 4.52) in DD ones, without differences in gender (Table 4).

### 3.6. Sociodemographic and Clinical Characteristics of Patients Attended during Lockdown Period

During the lockdown period 77 patients were attended by the CLAS and could not be included for interview assessment. No differences were found in relation to gender proportion and all the clinical and sociodemographic variables analysed in this study (Appendix A). Of these patients, 13 (16.8%) were diagnosed with COVID-19 infection.

## 4. Discussion

The prevalence of DD among patients with SUD admitted to the general hospital was around 37% and depression was the most frequent psychiatric disorder in both genders, representing more than a third of the sample. This prevalence is described for the first time in a CLAS of a general hospital. Other studies had reported a depression incidence of 10–15% in general hospital inpatients [27,28], but not in SUD patients.

Although in our study women tended to have more DD than men (53.8% vs. 32.7%), differences were not significant, probably related to their low number. This is in contrast with other studies, where women with SUD presented more DD than men [28,29]. Depression was the most common DD in both genders, while panic and generalized anxiety were more frequent in women. We could not confirm results of other studies [14,15,16,17,18], except for a higher prevalence of anxiety disorders in women compared to men. We think a possible explanation, besides the small sample size, could be an under-diagnosis or lower self-report of consumption, specifically in women, that limit their seeking consultation. 

Regarding the self-perceived QoL, there are several communications that associate worse QoL with the presence of addiction and other mental health problems [7,8,10]. In our sample there were no differences between patients with and without DD, when separating by gender; however, in women, the QoL self-perception was significantly worse, which could not be explained by other analysed sociodemographic and clinical factors. We observed that the presence of HIV antibodies was associated with more DD but not with worse QoL. QoL has been proposed as a neglected factor that could play a critical role in sustaining remission [30], according to these results; therefore, women with DD had a more difficult path to recovery.

In relation to addiction severity, there was a tendency for it to be worse in women and in patients with DD, although such differences were non-significant. 

The COVID-19 pandemic changed conditions for everybody, including patients and clinicians. We analysed the data of patients that could not be interviewed by the study researcher and observed no marked differences with respect to the other participants, with the exception of 13 diagnosed with COVID-19. Nevertheless, other factors, such as isolation and the infection itself, could have had different implications in the psychopathology and well being of these patients; therefore, it will be crucial to look forward prospectively. 

Regarding the limitations of the study, it should be noted that the sample was small, especially the number of women, which could be associated with some bias and limit external validity; women are usually underrepresented in addiction research and it is essential to design the projects with gender perspective. In addition, due to the COVID-19 pandemic, the interviews ceased for a few months, which also led to the final sample being smaller than expected. Nevertheless, we adapted to this situation by adding an additional objective. There were no differences in gender proportion or sociodemographic variables of included patients versus non-included ones; the sample therefore should be representative. In addition, the comorbid diagnosis has been obtained with a screening tool and not by a structured interview. For this reason, although there is a high sensitivity, there would be less specificity to obtain diagnosis; however, previous studies validating the DDSI screening tool have found acceptable specificity for the majority of diagnoses [23].

Patients with more severe clinical conditions were excluded as they were unable to complete the assessment, and also those diagnosed with COVID-19 infection. It is thus possible that the QoL and addiction severity scores might have been worse. 

In addition, there could have been a selection bias since only those patients that general medicine deemed necessary were assessed. There may have been others in which the SUD was considered less important, either because the patients did not report it, were abstinent, or it was simply not detected. This might also explain why the number of women with SUD was much lower than that of men [31]. Regarding women, it would also have been useful to obtain information about their backgrounds, for instance, if they were mothers, had suffered gender violence and/or sexual abuse and so on, and observe whether such factors were more prevalent in patients with DD. More stigma is associated with addicted women than men which could be a reason not to seek help during hospital admission.

## 5. Conclusions

In this study, patients were assessed by a psychiatry liaison service specializing in addiction. The study provides some insights into the characteristics of in-ward patients with an SUD and reinforces the need of an individualized approach.

Our findings suggest a high prevalence of DD amongst patients that were attended by the addiction liaison service of a general hospital. Furthermore, parameters such as QoL and addiction severity were worse when DD was present, especially in the group of women. Such results support the importance of routinely exploring substance consumption in hospitalized patients, and assessing the presence of dual pathologies, as they can play a role not only in medical pathology, but also in QoL, and particularly in women. More studies will be necessary to determine the implications of these differences in order to elucidate specific needs. 

## Figures and Tables

**Figure 1 jcm-10-05572-f001:**
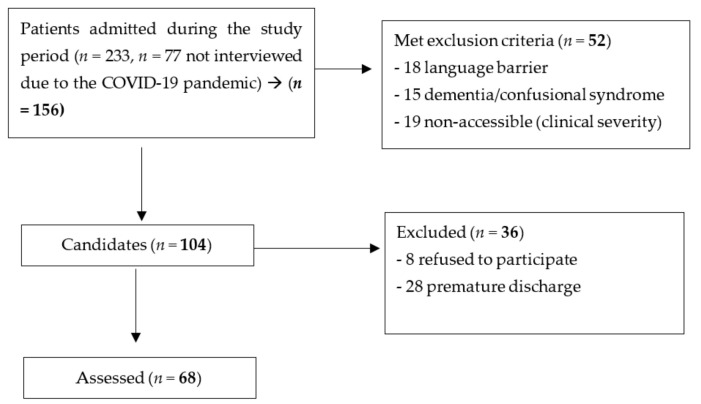
Flowchart for study enrolment.

**Table 1 jcm-10-05572-t001:** Demographic characteristics of the sample stratified by gender.

	Women *n* = 13	Men *n* = 55	Participants *n* = 68	
	*n* (%)	*n* (%)	*n* (%)	*p*
Age (mean ± SD) years	47.92 ± 10.15	51.71 ± 11.97	50.99 ± 11.67	0.310
Civil status				0.433
Single	5 (41.7)	23 (43.4)	28 (41.1)	
Married/partner	3 (25)	13 (24.5)	16 (23.5)	
Others	5 (38.4)	19 (34.5)	24 (35.2)	
Origin				0.217
National	11 (84.6)	37 (67.3)	48 (70.6)	
Employment situation				0.740
Employed	1 (8.3)	7 (12.7)	8 (11.7)	
Unemployed	8 (66.7)	25 (45.4)	33 (48.5)	
Retired	4 (25)	11 (20)	15 (22.1)	
Others	0	12 (21.8)	12 (17.6)	
Living with				0.336
Nobody	0	13 (23.6)	13 (19.7)	
Family	9 (69.2)	21 (38.1)	30 (45.5)	
Homeless	3 (23.1)	13 (23.6)	16 (24.2)	
Others	1 (7.7)	8 (15.5)	9 (13.2)	
Criminal records				0.559
No	8 (72.7)	35 (67.3)	43 (63.2)	

SD: Standard deviation.

**Table 2 jcm-10-05572-t002:** Clinical characteristics of the sample, stratified by gender.

		Women (*n* = 13)	Men (*n* = 55)	Participants (*n* = 68)	
		*n*%	*n*%	*n*%	*p*
Main drug:	Opiates	5 (38.5)	21 (38.2)	26 (38.2)	0.896
	Alcohol	6 (46.2)	26 (47.3)	32 (47.1)	
	Cocaine	2 (15.4)	5 (9.1)	7 (10.3)	
	Amphetamines	0	2 (3.6)	2 (3)	
	Tobacco	0	1 (1.8)	1 (1.5)	
Commencement age of main drug ($\bar{x}\;$± SD), years		23 ± 8.26	17.83 ± 5.91	18.71 ± 6.59	**0.018**
Total abstinence time ($\bar{x}$ ± SD), months		18.73 ± 24.50	25.31 ± 37.28	24.15 ± 35.27	0.903
Time since last consumption of the main drug ($\bar{x\;}$± SD), months		6.08 ± 20.76	3.98 ± 14.30	4.37 ± 15.54	0.850
Patients previously involved in an addiction treatment		10 (76.9)	30 (55.6)	40 (58.9)	0.159
Age at first addiction treatment ($\bar{x\;}$± SD) years		35.75 ± 13.26	34.39 ± 14.9	34.74 ± 14.29	0.572
HIV antibodies positive		3 (23.1)	10 (18.2)	13 (19.1)	0.702
Ab HCV serology positive		4 (30.8)	16 (29.1)	20 (29.4)	0.954
Ab core HBV serology positive		4 (30.8)	10 (18.2)	14 (20.6)	0.601
Ag surface HBV positive		0	2 (3.6)	2 (2.9)	0.728
Chronic liver disease		5 (38.5)	19 (34.5)	24 (35.3)	0.909

SD: standard deviation, SUD: substance use disorder, HIV: human immunodeficiency viruses, HBV: hepatitis B virus, HCV: hepatitis C virus, Ag. Antigen, Ab. Antibody. Bold numbers represent statistically significant results.

**Table 3 jcm-10-05572-t003:** Prevalence of DD amongst patients that completed the interview, stratified by gender.

	Women *n* = 13	Men *n* = 55	Participants *n* = 68	
**Psychiatric Diagnoses**	***n* (%)**	***n* (%)**	***n* (%)**	** *p* **
Dual Disorder	7 (53.8)	18 (32.7)	25 (36.8)	0.156
Panic	6 (46.2)	7 (12.7)	13 (19.1)	**0.019**
Generalized anxiety	5 (38.5)	6 (10.9)	11 (16.2)	**0.049**
Simple phobia	3 (23.1)	3 (5.5)	6 (8.8)	0.104
Social phobia	1 (7.7)	4 (7.3)	5 (7.4)	0.958
Agoraphobia	1 (7.7)	2 (3.6)	3 (4.4)	0.522
Dysthymia	3 (23.1)	4 (7.3)	7 (10.3)	0.092
Depression	6 (46.2)	17 (30.9)	23 (33.8)	0.296
Mania	3 (23.1)	8 (14.5)	11 (16.2)	0.174
Psychosis	4 (30.8)	9 (16.4)	13 (19.1)	0.445
ADHD	3 (23.1)	6 (10.9)	9 (13.2)	0.479
PTSD	1 (7.7)	8 (14.5)	9 (13.2)	0.512

DD: Dual Disorder, ADHD: attention deficit hyperactivity disorder, PTSD: post-traumatic stress disorder. Patients can present more than one psychiatric diagnosis. Bold numbers represent statistically significant results.

**Table 4 jcm-10-05572-t004:** Total scores of WHO and SDS in DD vs. non-DD patients.

	DD	Women (*n* = 13)	Men (*n* = 55)	Total (*n* = 68)	*p*
WHO ($\bar{x}$ ± SD)	No	62 ± 23.83	52.32 ± 30.9	53.67 ± 30.03	0.404
	Yes	21.71 ± 21.52	50 ± 31.02	42.08 ± 31.0	**0.020**
SDS ($\bar{x}$ ± SD)	No	4.5 ± 3.27	6.92 ± 4.14	6.58 ± 4.08	0.146
	Yes	10.14 ± 3.72	7.39 ± 4.65	8.16 ± 4.52	0.145

WHO: WHO well-being Index; SDS: Severity of Dependence Scale; SD: standard deviation. Bold numbers represent statistically significant results.

## Data Availability

Data are available upon request. Please contact the corresponding author if they are required.

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
