# Peer review of "Dual Disorders in the Consultation Liaison Addiction Service: Gender Perspective and Quality of Life"

_jcm, 2021, doi:10.3390/jcm10235572_

Round 1

Reviewer 1 Report

This paper presents dual disorders (DD) among patients with substance use disorder (SUD) admitted to a general hospital and attended by a Consultation Liaison Addiction Service (CLAS) and assessed its association with addiction severity and quality of life from a gender perspective. It considered to be a valuable paper comparing gender perspective and dual disorders with SUD in the context of COVID-19. Some comments and suggestions for this paper are as follows:

  • In general, It is a bit disappointing that the number of female patients is remarkably small (men: women, 13:55) that it is difficult to compare them.
  • Lines 53-56: The evidence or references for the sentence, “Women are more vulnerable ~ a greater degree of addiction severity”, are missing. The lack of evidence makes this sentence less scientific.
  • Line 142: In “Met exclusion criteria” of Figure 1, number of total patients and each case is not matched.
  • Some tables do not match each other or are incorrect. It is difficult to understand the reader by not writing the numbers properly. So please rewrite this part.
  • The number of “Living with others” is 7 out of 68 patients in Table 1, while the number of “Living with others” is 9 out of 68 patients in Supplementary material.
  • Line 19: (depression) men: 30.9%, p=0.296 / (panic) men: 12.7%
  • Reference [16] was not indicated.
  • In Table 2 & Table 6 in Supplementary material. So please check again.
  • Ab HVC Serology positiveè Ab HCV Serology positive
  • Ab core HVB Serology positiveè Ab core HBV serology positive
  • There is no HIV statistics, though there is a short explanation under table 2. Also I confused line 192~194, in Table 2. There is no HIV results.
  • I think it is a good study to reveal a high prevalence of DD amongst patients that were attended by the addiction liaison service of a general hospital and to show differences in life quality from gender perspective.
  • Although the number of female patients is small, the results of the study are commendable, especially, the quality of life in women with SUD was significantly lower than men with SUD. We hope that further studies on patients with DD will be continued in the future.

Thank you.

Reviewer 2 Report

Manuscript number: jcm-1442332

Dual Disorders in the Consultation Liaison Addiction Service: gender perspective and quality of life. A study conducted at the onset of the COVID-19 epidemic.

The manuscript aims at investigating the comorbidity of psychiatric disorders in a population of  patients with substance use disorders, with a focus on gender differences and quality of life.

The topic is current, clinically interesting, and appropriate for the Journal. The introduction is comprehensive and provides all the necessary premises, the methods are well written and reproducible, the results are clearly stated, and the conclusions are derived from the results. The major limitation, as specified by the authors, is the small sample size, especially for women. The major limitation, as specified by the authors, is the small sample, especially with regard to women, but I think it is still an interesting preliminary study, certainly to be extended and replicated.

I just have some minor comments:

  • In my opinion, the authors gave too much weight to the fact that the study was conducted during the Covid-19 epidemic. For example, in the title, I would remove this detail. I understand that they want to explain the small sample size, but it has no association with the results shown.
  • Psychiatric diagnoses refer to what period? Are they current or lifetime?
  • Limitations should state that psychiatric diagnoses were made through a screening tool and not through a structured clinical interview.
  • Table 4 lacks abbreviations at the bottom. Also missing in all tables is the note in the caption explaining what the bold values mean.
  • As a general comment, the English language needs a minor revision and the text needs to be double-checked for the presence of some typos.

Given the interest of the content and the appropriateness of the topic for the Journal of Clinical Medicine, I would consider the possibility publishing the article after some minor revision.
